# Designing Cellular Structures for Additive Manufacturing Using Voronoi–Monte Carlo Approach

**DOI:** 10.3390/polym11071158

**Published:** 2019-07-07

**Authors:** Tao Liu, Sofiane Guessasma, Jihong Zhu, Weihong Zhang

**Affiliations:** 1State IJR Center of Aerospace Design and Additive Manufacturing, School of Mechanical Engineering, Northwestern Polytechnical University, Xi’an 710072, China; 2UR1268 Biopolymères Interactions Assemblages, INRA, F-44300 Nantes, France

**Keywords:** Monte Carlo, Voronoi, stereolithography, mechanical performance, cellular structure

## Abstract

This study aims at reporting a strategy of designing cellular materials based on Voronoi–Monte Carlo approach for additive manufacturing. The approach is implemented to produce a fully connected cellular structure in the design space without producing material discontinuity. The main characteristics of the cellular structure, such as the density and the cell size, are controlled by means of two generation parameters, namely the number of seed points and the relaxation time. The generated cellular structures representing various designs of generated cellular wrenches are converted into surface tessellations and manufactured using stereolithography. Bending experiments are performed up to the rupture point and main attributes representing the performance of the SL-based cellular wrenches are studied with respect to the generation parameters. The results show only slight difference between CAD (Computer-Aided Design) models of the design and the real printed parts. The number of seed points is found to control the main feature of the wrench performance whereas the relaxation time is found to have a secondary effect.

## 1. Introduction

A cellular structure is a medium composed of solid and gaseous phases, and consists of an interconnected network of solid struts or plates that form the edges and faces of the cells [1]. According to the geometric configuration of each cell unit, the cellular structures can be categorized as foams, 2D lattice structures such as honeycombs, or 3D lattice structures [2]. Because of the numerous ways of controlling the position of the solid and air phases, the cellular structures exhibit more flexibility compared to other types of structures to adapt their architecture to the required functional property. Cellular materials are, for instance, used for applications where a high stiffness-weight ratio is needed [3]. They are also used as high heat rate dissipaters through active cooling [4], energy absorbers by inducing a large deformation at a relatively low stress level [5], and acoustic insulators thanks to the large number of internal pores [6]. Owing to its excellent physical performances, cellular structures have been widely used throughout ultralight structures [3], energy and noise absorbers [7,8], low thermal expansion structures [9], thermal insulators [10], conformal cooling designs [11], and substitutes for tissue engineering [12].

Conventional manufacturing methods such as foaming, powder metallurgy, sheet metal forming, or wire bonding processes [13] experience difficulties to control the internal microstructure of the cellular structures and fail to fabricate 3D lattice structures with complex cells. Manufacturing constraints of these fabrication techniques become a barrier to design more elaborated products allowing for a fine control of the performance. By fabricating parts layer by layer, additive manufacturing (AM) show a strong capacity in manufacturing parts with a high level of shape complexity. In practice, AM has been extensively applied to cellular structure fabrication [14]. Some of the most popular AM systems include stereolithography (SL), laser sintering (LS), fused deposition modelling (FDM) and laminated object manufacturing (LOM), which use liquid, filament/paste, powder and solid sheet material, respectively [15].

Stereolithography as one of the popular AM techniques exhibits a high fabrication accuracy. SL has achieved a remarkable success in manufacturing cellular structures for tissue engineering [16]. SL integrates several modules, including model slicing, laser devices control, material selection, machine parameter setup, and post-processing modules [17]. As a working technological solution, individual processes of SL are likely to introduce some geometric errors or defects. A large amount of the research work has been dedicated to improve the as-fabricated part quality by optimizing the SL process against known imperfections such as shape and dimension errors, stair casing, distortion and shrinkage phenomena appearing during the post-curing. For instance, Zhao and Luc [18] developed a practical approach for adaptive direct slicing of solid models based on the area deviation ratio. Direct slicing avoids an intermediate representation, and besides this, adaptive slicing is able to take into account the curvature of the surface by modifying the layer thickness, which is an effective way to alleviate the staircase effect. Zhou et al. [19] reported five most important input process parameters (the layer thickness, resultant overcure, hatch space, blade gap, and part location) that affect geometry accuracy and surface roughness with a leading influence of layer thickness and laser scanning duration. Canellidis et al. [20] employed a genetic algorithm to achieve the optimum build orientation in consideration of a multi-criteria objective function containing build time, the post-processing time, and the surface roughness. Recently, process monitoring and real-time process control are introduced in AM to improve the part quality and repeatability [21].

By means of manufacturing process optimization, AM is further improved to overcome its limitation and drawbacks. Thanks to the geometrical freedom provided by AM, the design space of cellular structures has been greatly enlarged. Structural optimization methods including size optimization, shape optimization, and topology optimization are used to design cellular structure. Compared with the other two methods, topology optimization method is able to change its topology to achieve an innovative design especially during concept design phase. This is a powerful free-form design tool, which has been widely applied to various engineering problems, and has achieved remarkable success in both theoretical research and practical applications [22,23]. For cellular structures, the topology of cells is extremely significant factor for structural macro performance, by designing new topologies or shapes of each unit cell, some unique desired properties can be achieved. Up to now, different types of topology optimization methods such as solid isotropic material penalization method (SIMP) [24,25], level set method [26], bidirectional evolutionary structural optimization method (BESO) [27] have been successfully developed to design a unit cell for desired mechanical properties such as the elastic modulus, shear modulus, and Poisson’s ratio. Besides, some optimization models are further developed to design cell topology and structure for thermal conductivity [28], electromagnetic property [29], and multifunctional characteristics [30,31].

This paper is aimed at designing structures with cellular structure by coupling Voronoi and Monte Carlo algorithms. Compared with topology optimization method, the Voronoi–Monte Carlo technique can be considered as a simple way to control the cell density, local distribution, and morphology. Indeed, according to the review paper by Giannitelli et al. [32], this type of stochastic generation approach can be meaningful to mimic the randomness of cellular structures in natural tissues such as bones. Multiple Voronoi tessellation algorithms exist such as the Poisson–Voronoi tessellation or the centroid Voronoi tessellation used to simulate the material microstructures [33]. These variants provide a realistic topological rendering by means of simple geometrical criteria, which are related to the controlled growth of seed points also called germs. In the field of additive manufacturing, Lee et al. [34] developed a Voronoi approach to design hollow structures without the need to input support material. The Monte Carlo approach is also considered by Rodgers et al. [35] to simulate the 3D grain structure in additively manufactured metals.

In this study, the combination of Voronoi–Monte Carlo approach is attempted as a filling pattern in additively manufactured structures to allow a controlled connectivity between the boundary edges or faces. A typical wrench structure is designed using this technique by varying the amount of seed points used to control the porosity content. Then, a Monte Carlo algorithm is introduced to generate the cell curvature. These designed structures are fabricated via SL. The volume mismatch between the designed and the as-fabricated parts is quantitatively analysed. Finally, bending experiments are implemented to study the effect of the generation parameters on the rendering of the 3D printed designs by considering the effect on the stiffness and maximum bearing capacity. The extension of the method to more complex 3D designs is also discussed.

## 2. Mathematical Background

The generation of a cellular structure within a 2D design is based on Voronoi–Monte Carlo approach. The first step is the space tessellation using the Voronoi algorithm. This is a cooperative generation approach that provides a subdivision of the design into a set of cells without producing discontinuities [36]. The geometrical characteristics of the cellular structure at the end of this step are only determined by the number of seed points *N_S_* that are randomly positioned within the design domain [37]. The 2D design domain considered in this study is an in-plane view of a wrench occupying a space defined by a grid of square points (Figure 1). The typical plane dimensions of the wrench are 186 mm × 16 mm. The initial state of the cellular structure can be defined as
(1)$$\begin{array}{c} x_{i}=\mathrm{random}\left(X\right)\\ y_{i}=\mathrm{random}\left(Y\right)\\ p\left(x_{i},y_{i}\right)=i;p\left(X,Y\right)\in \Omega ;i\in \left(1,N_{S}\right) \end{array}$$
where p is a label associated with an admissible point from the design space Ω, $x_{i}$ and $y_{i}$ are the coordinates of the seed point *i*.

In Figure 1, the image on the top represents a population of 200 seeds.

In this example, the seed distribution results in a surface density of 5.67 cm^−2^. According to the Voronoi approach, seed points are allowed to grow under a constant rate (second image from the top in Figure 1). When contact between the cells is detected, the growth is stopped and a boundary is set. This boundary corresponds to the thickness of the cell wall. This can be translated in a mathematical language using
(2)$$\begin{array}{c} p\left(x_{j},y_{j}\right)=i\\ \mathrm{if}\;\sqrt{{\left(x_{j}-x_{i}\right)}^{2}+{\left(y_{j}-y_{i}\right)}^{2}}\leq R_{i}\\ \mathrm{and}\;p\left(x_{j},y_{j}\right)=0 \end{array}$$
where $x_{j},y_{j}$ are the coordinates of a point in the design domain Ω, $R_{i}$ is the radius of the cell associated with the seed point i.

The process is repeated for all increments $\delta R_{i}$ of seed points until all points from the design domains are labelled
(3)$$R_{i}=R_{i}+\delta R_{i}$$

When all the seed points evolve to complete cells, the final space tessellation is achieved (third image from the top in Figure 1). Further modification of the cellular structure is considered by adding a Monte Carlo step, which allows the elimination of excess of virtual energy represented by the number of cell walls. This cell wall elimination occurs according to an algorithm adapted from the simulation of polycrystal growth. The excess of virtual cell wall energy E can be written as follows
(4)$$E=\left(E_{0}/2\right)\sum \left(1-\delta_{ij}\right)$$
where $E_{0}$ is the virtual interface energy associated with a unit cell wall in the design, δ is the kronecker function.

The change of energy state is decided based on Monte Carlo principle by issuing the probability that a given point $\left(x_{j},y_{j}\right)$ from a given cell i changes its label $p\left(x_{j},y_{j}\right)$, according to the probability
(5)$$P=exp\left(-\Delta E_{ik}/kT\right)$$
where $\Delta E_{ik}$ is the change in cell wall energy associated to a state change of a given point $\left(x_{j},y_{j}\right)$ from i to k. This also means that this point switches into another cell label, kT is a positive constant that refers to the virtual thermal energy.

This process adds some curvature to the cell wall structure in order to meet the topological requirement of equilibrium (image on the bottom in Figure 1). The number of attempts to switch the state of all points in the design defines one Monte Carlo step. The number of Monte Carlo steps $t_{R}$ refers to the relaxation time, which is here considered as a stopping criterion. When this time is large enough, the process leads to the disappearance of all cell walls within the design domain.

Thus, the main variables of the generation process using Voronoi/Monte Carlo approaches are the number of seed points $N_{S}$, and the number of Monte Carlo steps $t_{R}$.

Four levels are used for the number of seed points (50, 100, 200, 500). Knowing that the design domain has a typical surface area of 35.26 cm^2^, the surface density of the seed points varies from 1.42 to 14.18 cm^−2^. The number of seed points is combined with two levels of relaxation time $t_{R}$ namely 71 MCS and 100 MCS. Thus, the total number of studied configurations is eight. Because of the randomness of the Monte Carlo process, each configuration is repeated four times. This makes the total number of designs equal to 32.

## 3. Experimental Layout

The 2D cellular structures representing the geometry of the wrench are converted into 3D designs for further processing using stereolithography (Figure 2a). For this purpose, an extrusion through the thickness of the design by 6 mm is performed.

The main steps are the conversion of the CAD models into STL (Standardized Triangular Language) files, generation of the support structure, and the slicing of the designs into horizontal cross-sections with a resolution of 0.1 mm, which corresponds to the thickness of each slice (Figure 2a). It is worth mentioning that the support used for mechanical testing of the wrench is also added at this stage. All these steps are performed under the RpData environment (Hengtong Ltd., Xi’an, China). The 3D printing process is conducted on the SPS350B industrial machine purchased from Hengtong Ltd., Xi’an, China (Figure 2b). The process comprises the use of a laser source (beam diameter = 0.15 mm, UV wavelength = 355 nm, laser power = 220 mw, max scanning speed = 8 m/s). The laser draws a series of patterns corresponding to the frontiers in each slice at the surface of a vat, which is filled with a photosensitive resin (Figure 2b). The main physical properties of the resin in the liquid and solid states are given in Table 1.

After a solid layer is formed at the surface, the platform is displaced vertically to expose the liquid resin to the laser beam and the process continues until the full thickness of the wrench is built. The solid polymeric structure in its green state is further processed by an ultrasonic cleaning step followed by an exposure to UV light for 16 min as a post curing step (Figure 2b).

Figure 3 shows the 3D printed wrenches processed using different generation conditions. Depending on the density of seed points, the 3D printed wrenches conform more or less to the definition of a cellular structure.

The lack of airiness for $N_{S}$ = 500 suggests more a porous wrench than a typical cellular structure. A total of 32 specimens are printed considering the replications and the levels for each generation parameter. To this number is added the control group consisting of two solid wrenches without any cells (Figure 3c).

The solid and cellular wrenches are subjected to bending testing using an electromechanical universal test machine from Testresources Inc. (Shakopee, MN, USA) equipped with a load cell of 100 kN (Figure 4). Testing is conducted until the rupture of the wrench or the support. The moving arm is displaced with a constant rate of 10 mm/min. Optical recording is performed during the wrench deformation using a CMOS camera (Fastcam SA-X2 from Photron company, Tokyo, Japan). The acquisition is performed using a resolution of 2448 × 2048 pixels (pixel size is 71.43 µm) and a frame rate of 50 fps (frame per second).

## 4. Results and Discussion

### 4.1. Voronoi/Monte Carlo Based Cellular Wrench

The cellular wrenches depicted in Figure 3 differ substantially in term of cell density and morphology. According to the Voronoi–Monte Carlo generation scheme, the number of seed points controls the size of the cells within the wrenches. For the control group, the number of seed points is considered as infinite, which results in the fully solid wrench as shown in Figure 3c. For the remaining cellular wrenches, some imperfections can be expected, such as trapped resin within the cells especially when the density of seed points increases. The relaxation time has also an equivalent effect on the density of cells because a short relaxation time does not allow a large elimination of cell walls and thus the cell size remains small. In addition to this effect on the cell size, the morphology of the cells and their connectivity strongly depend on this parameter.

It is known that the mechanical performance of a cellular structure is mainly dependent on its volume fraction. In order to quantify the difference between the CAD models and the real parts, the volumes of the solid phase in the real and virtual wrenches are measured for all generation parameters. Figure 5 shows the relative variation of the solid phase volume as a function of the number of seed points for the two considered relaxation times. This relative change in volume ΔV is defined as
(6)$$\Delta V=100\times \left(V_{PRT}-V_{CAD}\right)/V_{CAD}$$
where $V_{CAD}$ and $V_{PRT}$ are the volumes of the CAD model and the part manufactured using stereolithography.

For the control group composed of the fully solid wrenches, the volume mismatch is 0.85 ± 0.35%. Figure 5 shows that the volume fraction of the solid within the cellular wrenches ($V_{f}$) is nonlinearly dependent on the number of seed points.

The small error bars suggest a minor effect of the randomness of the generations. When the relaxation time is increased from 71 MCS to 100 MCS, the process of transforming the straight cell walls into curved ones is concomitant with increasing the cell size. This process lowers the volume fraction of the solid as shown in Figure 5. As for the relative change in volume (ΔV), it follows the same nonlinear trend with the number of seed points. This tendency is justified by the fact that small cells tend to be filled with resin resulting in volume mismatch as large as 3% for $N_{S}=500$. It has to be mentioned that the mismatch generated by the processing of the cellular wrenches ($N_{S}=100$) can be as small as the one generated for the fully solid wrenches.

It can be concluded that a high manufacturing accuracy is expected with cellular wrenches of low density (small number of seed points), and it remains acceptable for wrenches containing a small porosity content. The difference is rather small, which means that the printed wrenches are almost the same as the corresponding CAD models.

Typical deformation sequences of 3D printed cellular wrenches are presented in Figure 6. The four replicates of wrenches with $N_{S}$ = 100 and $t_{R}$ = 71 MCS are depicted in Figure 6a.

Since the wrench head experiences large bending stress compared to the handle, fracture is very likely to occur around this spot. But the specific position of fracture is different because the local cell wall distribution varies depending on the random positioning of the seed points. For some configurations, torsional deformation is observed (fourth replicate in Figure 6a), which is caused by the mechanical instabilities related to the cell collapsing. Figure 6b,c show the deformation sequences of cellular wrenches for an increasing number of seed points combined with short and long Monte Carlo relaxations, respectively. As can be seen, when the number of seed points increases up to 500, the fracture appears on the fixed support for cellular wrenches designed using a short relaxation time (Figure 6b). The achieved fracture pattern reflects a load bearing capabilities because of the low porosity content. When the relaxation time is increased, the same fracture pattern of the support is observed for a larger porosity content (Figure 6b) indicating a better load transfer between the cell walls (*N_S_* = 200 and *t_R_* = 100). This can be interpreted as a Monte Carlo relaxation that enforces a better geometrical equilibrium at the nodes (cell walls meeting at an average angle of 120° with three cell walls per node).

Figure 7 depicts the mechanical response of the cellular wrenches as a function of the number of seed points. Figure 7a also shows the bending response of the fully solid wrenches provided as a control group. The bending response of the solid wrenches can be stated as an elastic–plastic response with a brittle-like failure. This last property is known for resins that have been cured using UV light. The slope of the linear part is 9.1 ± 0.3 N/mm, whereas the maximum force achieved within the nonlinear part is 190 ± 7 N. The maximum displacement at the break point is 25 ± 1 mm.

Figure 7b shows the bending response of four replicates from the cellular wrenches using the generation parameters ($N_{S}$ = 50; $t_{R}$ = 71 MCS). Despite the apparent variability in terms of slope, maximum force, and displacement at break, the bending response of the cellular wrench has a parabolic shape comprising an elastic-plastic stage followed by smooth decrease of the bending force due to a slow degradation of the deformed wrench. The maximum force observed in the best case (replicate R4) represents 62 ± 3% of the maximum force obtained for the solid wrench. However, if the rupture properties are compared, the design of cellular wrenches allows more ductility and the displacement prior break doubles. Figure 7c depicts the response bending for an increasing number of seed points and a small relaxation time ($t_{R}$ = 71 MCS). The ranking of the wrench response significantly depends on the volume fraction of solid ($V_{f}$), which is tunable by means of the number of seed points ($N_{S}$). Both the maximum fore and the slope of the first linear part increase with the increase of the number of seed points. These two parameters are used hereafter to quantity the effect of process conditions on the performance of the 3D printed cellular wrenches. It has to be mentioned that the difference between the maximum forces for the cellular and the solid wrenches is only 13% in the best case ($N_{S}=500$). With this reduction in the difference between maximum forces, the tendency back to brittle like-failure is confirmed.

Figure 7d depicts the bending response for the same number of seed points but with a larger relaxation time ($t_{R}$ = 100 MCS). Similarly, the same ranking of the wrench performance is achieved based on the number of seed points but there seems to be slight differences. For instance, the curves for $N_{S}$ = 100 and $N_{S}$ = 200 are very close. The corresponding wrenches have almost the same slope. According to Figure 5, the volume fraction of the solid phase ($V_{f}$) is dominated by the number of seed points, and the wrench with 100 seed points has less material than the one with 200 seed points. But these two structures possess the same stiffness. This statement suggests that the distribution of seed point plays a considerable role in structure performance. The distribution of seed points allows an additional degree of freedom to design cellular structures with contrasted performance.

In order to validate this statement, the slope and maximum force of the four replicates are examined in Figure 8 for all printing conditions. It can be seen that the leading effect of the number of seed point is confirmed. However, the randomness of the distribution does not allow a reproducible behaviour as the tendencies for both the slope and the maximum force of all replicates are significantly different. It can be also argued that Monte Carlo relaxation adds more complexity in the final result as the dispersion obtained is significant between $t_{R}$ = 71 MCS and $t_{R}$ = 100 MCS.

Figure 9 summarizes the main extracted mechanical parameters from the bending test: slope, maximum bending force and maximum displacement as a function of the number of seed points *N_S_* and relaxation times *t_R_*. Since the seed points are introduced randomly, and the distribution of seed points has a considerable effect on structure performance, the results of the four replicates exhibit a dispersion from 5% to 23% for the slope, from 4% to 17% for the maximum force, and from 7% to 24% for the maximum displacement. When the number of seed points is larger than 200, the performance tendency is marked by a steady-state regime. This is only valid for the slope and the maximum force (Figure 9a).

The effect of relaxation time is negligible, considering the large fluctuation of the performance with respect to the random positioning of the seed points in the cellular wrench. The gain in performance represents 30% and 40% for the slope and the maximum force when the number of seed points is increased from 50 to 500. A loss in performance is observed in the case of the maximum displacement due to the lack of airiness when the number of seed points is increased in the full range. Such a loss represents 60%.

In order to check if the results shown in Figure 7, Figure 8 and Figure 9 can be further extended to other types of resins used in stereolithography, analysis of the physical properties of these resins should be undertaken. This analysis should consider the nature (UV versus thermal) and the level (duration versus intensity) of curing because as shown by Hague et al. [38], significant differences in tensile strength and elongation at break can be observed. The most admitted effect is the increase of the strength and stiffness and decrease of the elongation at break when the curing time is increased. Based on the results shown in Figure 9, the slope S of the bending response can be easily related to the stiffness of the resin. The correlation between the slope and the number of seed point can be approximated using a sigmoid-type function.
(7)$$S\left(N/{\mathrm{mm}}^{2}\right)=6.90-2.06\times exp\left(-\left(N_{S}-41\right)/34\right),\;R^{2}=0.99;\;t_{R}=71\;MCS$$
and
(8)$$S\left(N/{\mathrm{mm}}^{2}\right)=6.90-1.98\times exp\left(-\left(N_{S}-40\right)/32\right);\;R^{2}=0.98;\;t_{R}=100\;MCS$$

From Figure 5, the nonlinear correlation between the volume of the solid phase $V_{f}$ and the number of seed points can be exploited to relate the relative density ρ of the resin to the slope. Replacing $N_{S}$ in the former expressions by its equivalent in terms of $V_{f}$ leads to the following simplified function:
(9)$$S\left(N/{\mathrm{mm}}^{2}\right)=7.35\times \rho^{0.43};\;R^{2}=0.99;\;t_{R}=71\;MCS$$
and
(10)$$S\left(N/{\mathrm{mm}}^{2}\right)=7.82\times \rho^{0.56};\;R^{2}=0.99;\;t_{R}=100\;MCS$$
where $\rho =V_{f}/100$.

From these two expressions, the effect of Young’s modulus of the resin $E_{Y}$ can be introduced through the well-known expression for cellular materials that relates the relative density to its stiffness
(11)$$E_{Y}\left(N/{\mathrm{mm}}^{2}\right)=c\times \rho^{n}$$
where *c* and *n* are the prefactor and the exponent dependent on the type of the cellular structure.

According to a former work [39], the values of the prefactor and the exponent for the type cellular structure generated by Voronoi–Monte Carlo approach are as follows
(12)n≈3.0;c≈1.25
by replacing these in the expressions of the slope
(13)$$S\left(N/{\mathrm{mm}}^{2}\right)=5.88\times \left(E_{Y}/\rho^{2.57}\right);\;t_{R}=71\;MCS$$
and
(14)$$S\left(N/{\mathrm{mm}}^{2}\right)=6.26\times \left(E_{Y}/\rho^{2.44}\right);\;t_{R}=100\;MCS$$

From these last two expressions, the trends shown in Figure 9a are expected to evolve nonlinearly with the stiffness of the resin. If the mid-value of the tensile modulus of the resin used in this study (Table 1) is taken as a referential ($E_{Y}$ = 2.29 GPa), the expected change in the slope ΔS for another resin considered in [38] ($E_{Y}$ = 2.6 GPa) is as follows
(15)$$\Delta S\left(N/{\mathrm{mm}}^{2}\right)=5.88\times \left(0.31/\rho^{2.57}\right);\;t_{R}=71\;MCS$$
and
(16)$$\Delta S\left(N/{\mathrm{mm}}^{2}\right)=6.26\times \left(0.31/\rho^{2.44}\right);\;t_{R}=100\;MCS$$

Due to the large dispersion of the properties of resins used for stereolithography, the effect on the bending response can be substantial. For instance, the range of resin stiffness can as large as (0.78–3.3) GPa according to various published works [40,41]. It is worth mentioning that the same nonlinearity can be expected for the maximum force if this quantity is related to the strength properties of the cellular material according to the same scheme introduced in expression (11).

### 4.2. Extension to 3D Voronoi–Monte Carlo Scheme

In order to prove the relevance of the Voronoi/Monte Carlo scheme to generate more complex 3D structures, the expression (1) is modified to account for the possible generation of seed points in all space directions.
(17)$$\begin{array}{c} x_{i}=\mathrm{random}\left(X\right)\\ y_{i}=\mathrm{random}\left(Y\right)\\ z_{i}=\mathrm{random}\left(Z\right) \end{array}$$
where
(18)$$p\left(x_{i},y_{i},z_{i}\right)=i;p\left(X,Y,Z\right)\in \Omega ;i\in \left(1,N_{S}\right)$$

Figure 10 shows an example of spherical domain in which a Voronoi-like structure (Figure 10a) is generated using 250 seed points. The resulting cellular structure (Figure 10b) represents all the boundaries between at least three different sites or two sites plus an external boundary (the external shell in the case of a spherical domain). Further modification of the Voronoi tessellation result in more curved boundaries as shown in Figure 10d. The Monte Carlo process, in this case, is limited to a duration of 40 MCS, for which only slight change in the cell size can be expected. Using the same scheme for cellular structure generation, the open structure shown in Figure 10d differs from the one in Figure 10b by the additional curvature related to the Monte Carlo growth process.

Figure 11 shows the use of the exclusion concept in some parts of the design domain. By referring to the former example of spherical shape domain, three regions, where solid structures are needed, are excluded. This means that the initial position of the seed points is attempted elsewhere. Figure 11a shows the final design after a Monte Carlo process performed with 100 seed points. The selection of the cellular structure differs in this example with respect to Figure 10b as the boundaries of the cells are selected if at least these separate two cells. This means that the final structure in Figure 11b contains in the design domain close-cell structure by opposition to the open-cell structure in Figure 11b. Further growth of the cells (Figure 11c) by a large amount of time (70 MCS in this example) allows the formation of larger curved cells. The conversion of the design into a cellular structure based on two-neighbor boundary cell walls form a bending dominant structure (Figure 11d).

Figure 12 shows the slicing step of the cellular structure in Figure 11d prior printing using stereolithography, where the scaling is adjusted to obtain a part diameter of 40 mm. The cross-section at depths 16 and 32 mm illustrates the requirement for internal support material to withstand the formation of the cellular structure. At the depths of 14 and 30 mm, the curvature of the cells generated from the Monte Carlo process is clearly depicted. At these particular depths that mark the limit between the cellular structure and the solid part, there is no need for an internal support material. The manufacturing step and the final rendering of the structures are illustrated in Figure 11b,d.

The manufacturing time for both structures is 137 min. The volume of the solid part is 19.98 cm^3^ and the cellular structure built using the Voronoi–Monte Carlo scheme exhibits a relative density of 0.60. The volume of the support material is 8.77 cm^3^, which represents nearly 44% of the solid part (Figure 13).

The generation of cellular structures based on Voronoi–Monte Carlo approach can be also extended to the design of technical parts for the purpose of weight saving (Figure 14). The definition of the design domain from a typical stl file can be performed, as shown in Figure 14, by excluding some regions where fully solid material is needed. The regions to be excluded can be decided based on mechanical criteria issued, for instance, from finite element computation. In the example shown in Figure 14a, these excluded regions correspond to the joining blocs. The generation of the Voronoi-like structure in Figure 14a is performed with 100 seed points. The smaller the number of seed points, the larger is the weight saving performance in the central part. Further development of the structure at the Monte Carlo stage allows also the control of the weight saving process by increasing the number of Monte Carlo steps as shown in Figure 14a. At a characteristic time of 100 MCS, more cells are eliminated by the cell growth process compared to the time of 20 MCS. The corresponding the cellular structures in Figure 14b are generated by a two-neighbor connectivity scheme. This scheme maintains a strong connectivity between the joining blocs even with a reduced number of seed points or with a long Monte Carlo stage. The Voronoi-based and the Monte Carlo–Voronoi based cellular designs offer different types of load bearing capabilities ranging from bending dominant to uniaxial dominant behaviour. Figure 14c shows the slicing step for both Voronoi-like and Voronoi–Monte Carlo-like cellular parts. The solid volume for these two parts is 17 cm^3^ and 14 cm^3^, respectively. The manufacturing requires a support volume of 2.41 cm^3^. This support material represents only 14%, and 16% of the solid volume for Voronoi-like and Voronoi–Monte Carlo-like cellular parts, respectively.

Figure 14d exhibits the rendering of the two printed parts. These are printed under a time of 2 h and most of the support material is removed after cleaning.

## 5. Conclusions

This work concludes that stereolithography is an adequate processing route to design cellular structures from a Voronoi–Monte Carlo generation scheme. By only means of two generation parameters, namely the number of seed points and the relaxation time, a wide variety of performance are achieved. The 3D printed cellular wrenches exhibit a wide variability of density and spatial distribution of the cells, which result in a versatile way to tune the bending performance especially the maximum force. The cellular wrenches introduce, for instance, more ductility compared to fully solid wrenches, where the displacement at break can be doubled without losing much of the bending resistance. The difference between the CAD models and the real parts is acceptable considering the 4% of mismatch obtained in the worst design case. The Voronoi–Monte Carlo scheme proves to be a robust way to design 3D printed cellular structures within complex design space while maintaining a full connectivity between the solid features in the three-dimensional space. The main advantage of the developed Vornoi–Monte Carlo approach combines the benefit of each algorithm. The Voronoi algorithm allows a tessellation of the space even for a complex design domain containing excluded regions without producing material discontinuities. The addition of the Monte Carlo step allows the continuity of the structure but also introduces curvature to the cell walls. This can be useful to design bending dominant cellular structures. In addition, adjustment of the cell size and the control of the bending capabilities are possible by varying the number of Monte Carlo steps.

The design of cellular structures using the Voronoi–Monte Carlo can be used in a more generic way to balance the brittleness of the cured resin known in stereolithography. The extension of the generation approach to the design of more complex 3D structures is straightforward and allows varieties of close-cell and open-cell filling forms. In the case of technical parts, weight saving performance can be controlled at both the Voronoi and the Monte Carlo stages allowing at the same time a tuning of the part mechanical response.

## Figures and Tables

**Figure 1 polymers-11-01158-f001:**
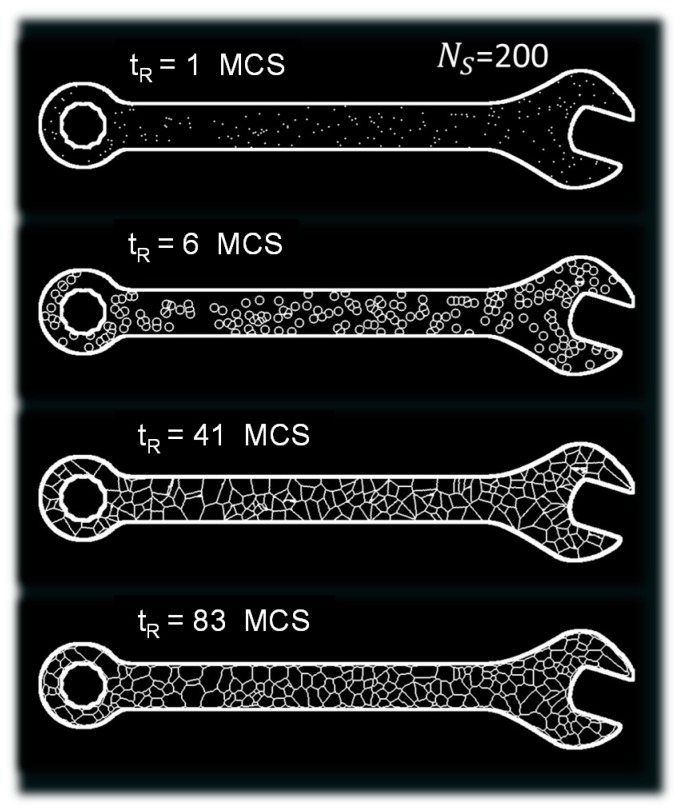
Evolution of the cellular structure within the design domain as a function of Monte Carlo steps for a fixed number of seeds (*N_S_* = 200).

**Figure 2 polymers-11-01158-f002:**
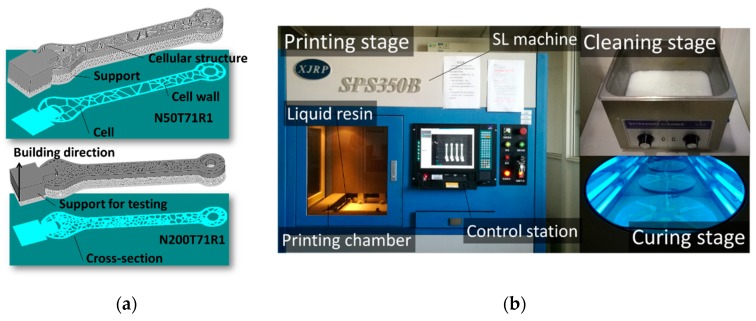
(**a**) Example of two CAD (Computer-Aided Design) models representing cellular designs of a wrench and their conversion into sliced objects for stereolithography. Sample nomenclature includes number of seed points *N*, relaxation time T and replicate number R, (**b**) Main steps of manufacturing using stereolithography equipment.

**Figure 3 polymers-11-01158-f003:**
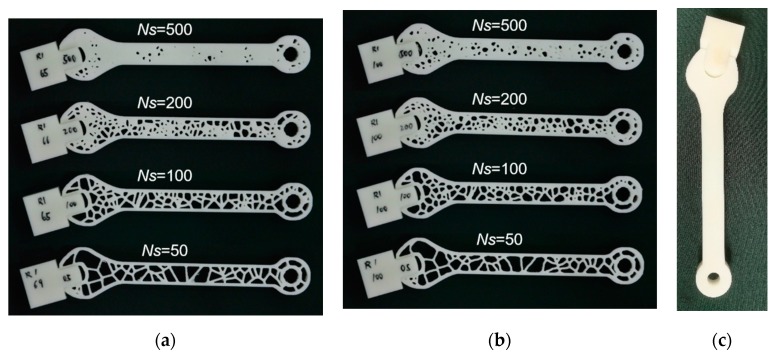
Cellular wrenches generated using Voronoi–Monte Carlo approach and manufactured using stereolithography for an increasing number of seed points $N_{S}$ and as a function of the relaxation time $t_{R}$ (**a**) 71 MCS, (**b**) 100 MCS, (**c**) fully solid wrench used as a control group.

**Figure 4 polymers-11-01158-f004:**
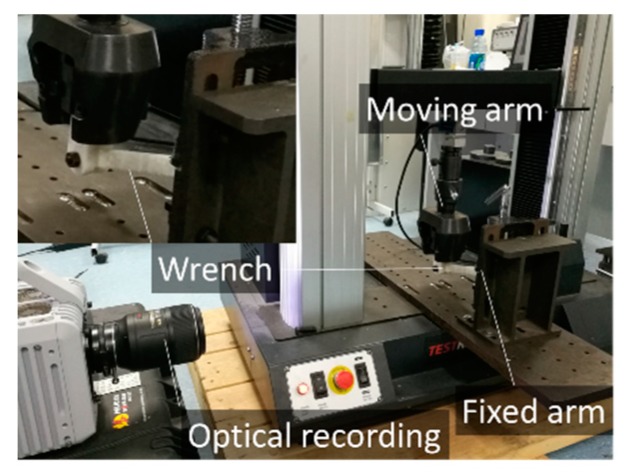
Experimental setup used to test the 3D printed cellular wrenches.

**Figure 5 polymers-11-01158-f005:**
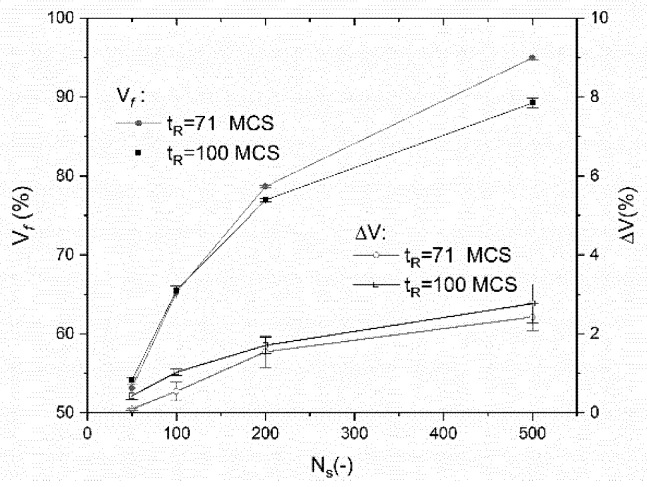
Measured volume (*V_f_*) of 3D printed cellular wrenches and comparison between the CAD and the real wrenches expressed as a relative variation of the solid phase volume (Δ*V*) as a function of the number of seed points ($N_{S}$) and the relaxation time ($t_{R}$).

**Figure 6 polymers-11-01158-f006:**
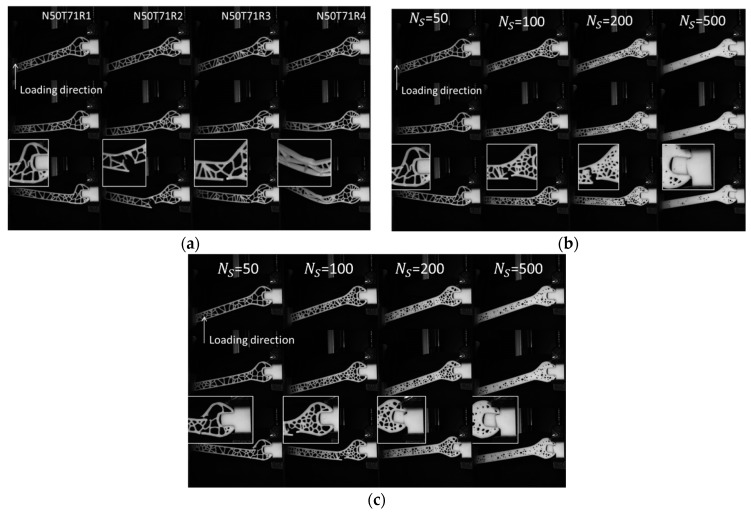
Deformation sequences of 3D printed cellular wrenches for (**a**) different replicates, (**b**) an increasing number of seed points ($t_{R}$ = 71 MCS), and for (**c**) a large relaxation time ($t_{R}$ = 100 MCS).

**Figure 7 polymers-11-01158-f007:**
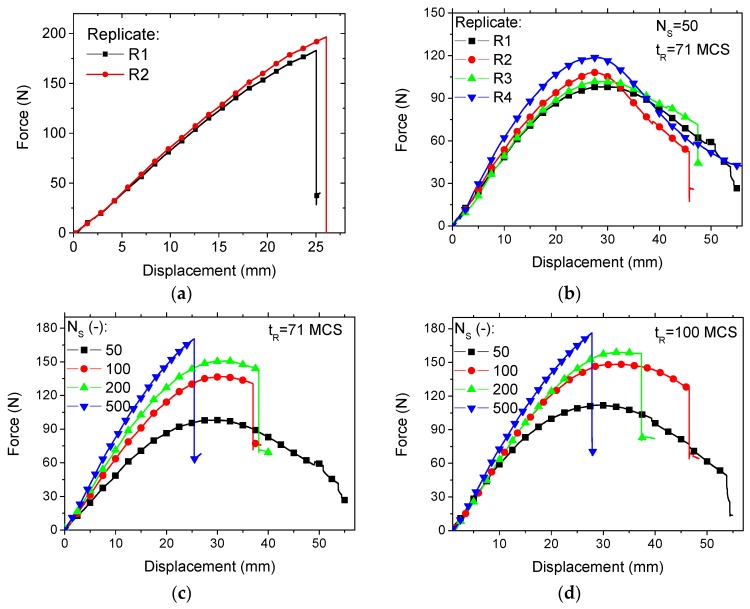
Result of bending experiment of 3D printed (**a**) fully solid and (**b**–**d**) cellular wrenches expressed as force–displacement curves for different (**b**) replicates and varied number of seed points *N_S_* combined with two relaxation times *t_R_*, (**c**) 71 MCS, (**d**) 100 MCS.

**Figure 8 polymers-11-01158-f008:**
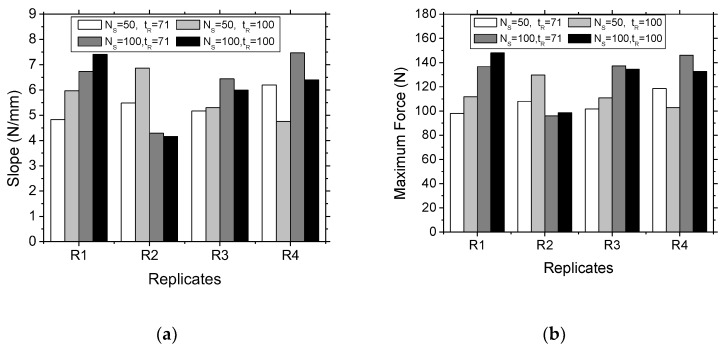
Influence of the randomness in Voronoi–Monte Carlo generation scheme on the measured (**a**) slope and, (**b**) maximum bending force of printed cellular wrenches for two different populations of seed points ($N_{S}$ = 50) and ($N_{S}$ = 100).

**Figure 9 polymers-11-01158-f009:**
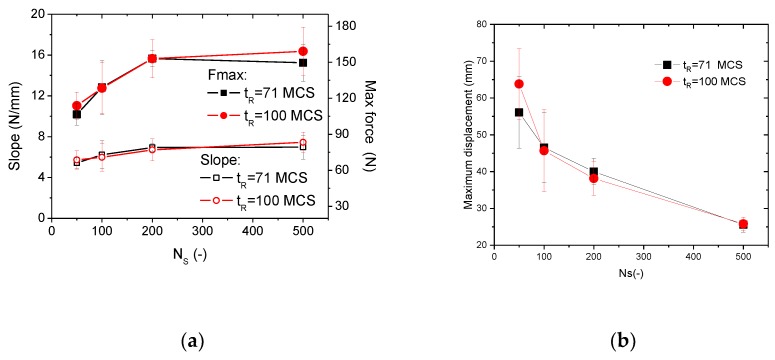
Main extracted mechanical parameters from the bending test: (**a**) slope and maximum force, (**b**) displacement at break as a function of the number of seed points *N_S_* and relaxation times *t_R_*.

**Figure 10 polymers-11-01158-f010:**
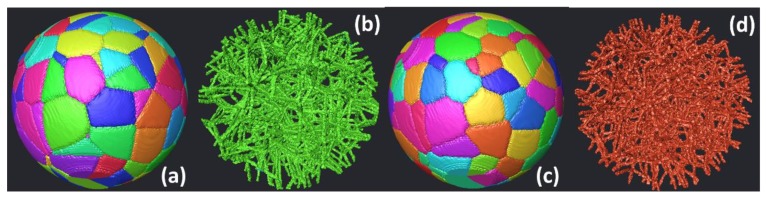
3D complex cellular structure generated using Voronoi–Monte Carlo scheme (**a**) Voronoi structure, (**b**) cellular network based on Voronoi open-cell boundaries, (**c**) Voronoi–Monte Carlo structure after 40 MCS, and corresponding (**d**) cellular structure.

**Figure 11 polymers-11-01158-f011:**
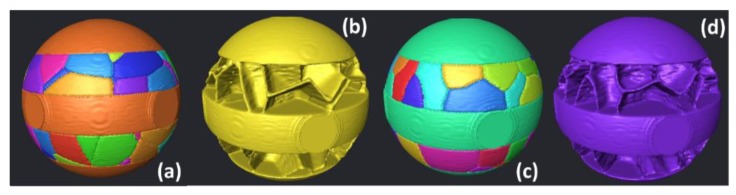
Generation of a 3D cellular structure containing excluded regions using Voronoi–Monte Carlo scheme (**a**) Voronoi structure, (**b**) cellular network based on Voronoi close-cell boundaries, (**c**) Voronoi–Monte Carlo structure after 70 MCS, and corresponding (**d**) cellular structure.

**Figure 12 polymers-11-01158-f012:**
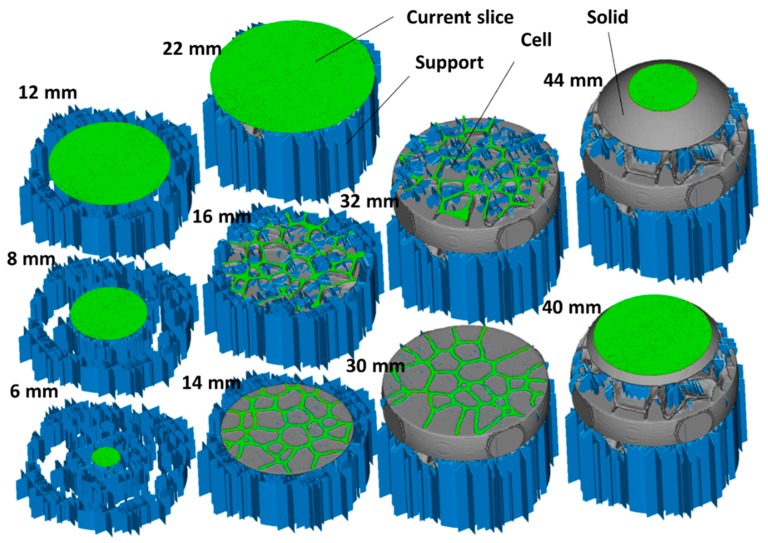
Slicing step showing different depth views of the Voronoi–Monte Carlo cellular structure prior manufacturing using stereolithography.

**Figure 13 polymers-11-01158-f013:**
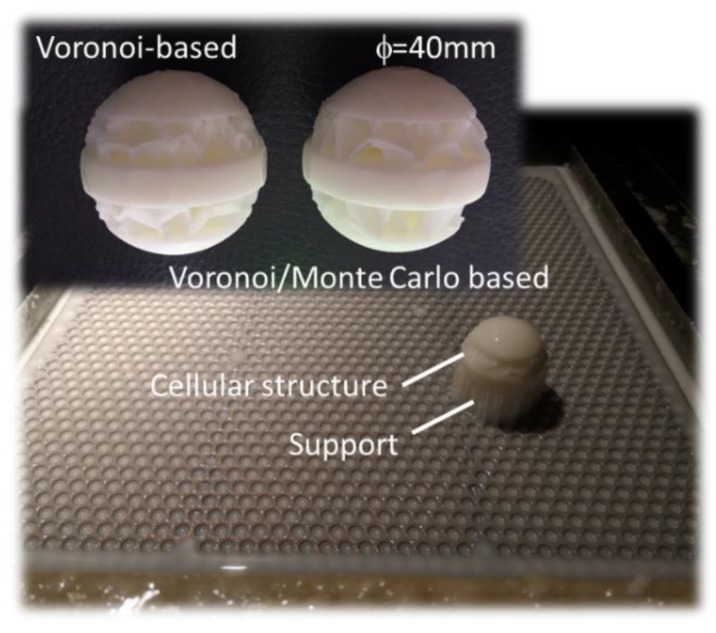
Rendering of the cellular structures designed using Voronoi–Monte Carlo generation scheme and manufactured using stereolithography.

**Figure 14 polymers-11-01158-f014:**
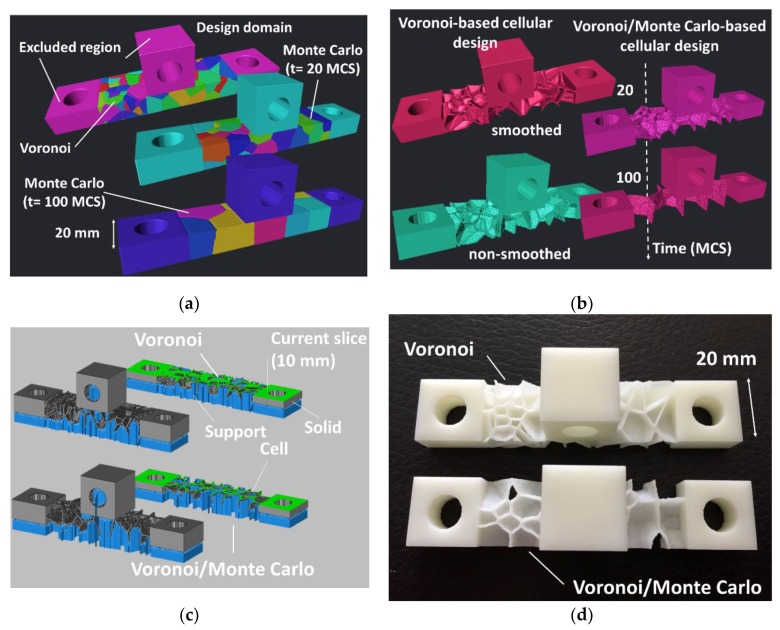
Extension of the Voronoi–Monte Carlo approach to the design of technical parts, (**a**) The generated Voronoi 3D structure based on 100 seed points and subsequent Monte Carlo structures at two different time steps (MCS), (**b**) the cellular designs achieved at different stages including the Voronoi-based and the Voronoi/Monte Carlo-based, (**c**) slicing and (**d**) 3D printed designs using stereolithography.

**Table 1 polymers-11-01158-t001:** Properties of the photosensitive resin at the liquid and solid states.

Liquid State	Magnitude	Solid State	Magnitude
Appearance	White	Flexural modulus	2692–2775 MPa
Density	1.13 g/cm^3^	Flexural strength	69–74 MPa
Viscosity	355 cps	Tensile modulus	2189–2395 MPa
Penetration distance	0.145 mm	Tensile strength	27–31 MPa
Critical exposure energy	9.3 mJ/cm^2^	Elongation at break	12–20%
Building layer thickness	0.1 mm	Density	1.16 g/cm^3^
-	-	Glass transition	62 °C

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
