# Peer review of "Designing Cellular Structures for Additive Manufacturing Using Voronoi–Monte Carlo Approach"

_polymers, 2019, doi:10.3390/polym11071158_

Reviewer 1 Report

A strategy of cellular structure design was reported, in which Voronoi-Monte Carlo has been used to create random patterns. The designed structures were constructed by stereolithography. The mechanical properties were tested to show variable performances. Therefore, the work could be acceptable for publication, after the authors address to the following concerns.

1)      Authors could consider adding a control group of a solid structure without any cells.

2)      What’s the material for the printed objectives? Is the trend of designs variable if other resins were used to build samples?

3)      Only 2D patterns have been discussed in the paper? What will happen if more complicated 3D patterns of cells were involved?

Author Response

Unité de Recherche

Biopolymères, Interactions, Assemblages

Nantes, June 14th 2019

Dear Editor,

You will find enclosed our revised manuscript entitled « Designing cellular structures for additive manufacturing using Voronoi-Monte Carlo approach »

and written by

Tao Liu, Sofiane Guessasma, Jihong Zhu, Weihong Zhang

 We would like to take this opportunity to thank the editor and the reviewers for their comments and suggestions that all helped to elaborate a new strong version of this work. You will find hereafter a detailed answer to all these comments. All changes are also identified in the manuscript in yellow colour. 

 Reviewer 1 :

Comments and Suggestions for Authors

A strategy of cellular structure design was reported, in which Voronoi-Monte Carlo has been used to create random patterns. The designed structures were constructed by stereolithography. The mechanical properties were tested to show variable performances. Therefore, the work could be acceptable for publication, after the authors address to the following concerns.

1)      Authors could consider adding a control group of a solid structure without any cells.

The control group of solid structures has been added and testing results are added and discussed in the new version. Fig. 1 shows an example of tested structure. The determination and discussion of the volume mismatch is introduced in page 6 :” For the control group composed...” + “It has to be mentioned that the...”. In addition, the discussion of the mechanical response is provided through the new Figure 7a and in page 7:” Also is shown in Figure 7a the bending response of the fully solid wrenches...” and in multiple places in pages 7-8 + in the conclusion section.

Some related amendments:

Page 5: “For the control group, the number of seed...

Figure 3c added + discussion in the text in page 5: “To this number is added the control group… without any cells (Figure 3a).”

Fig. 1. One of the solid wrenches manufactured to serve as a control group.

2)      What’s the material for the printed objectives? Is the trend of designs variable if other resins were used to build samples?

We introduced more information about the photosensitive resin. Physical data are summarised in Table 1. The equipment used for 3D printing works with only one type of resin. So, we could not use other resins to check what changes this would generate on the trends. In order to satisfy the comment of the reviewer, we analysed the literature work and provided a qualitative comparison based on the materials data, more particularly the stiffness. The strategy that we adopted is to rescale the engineering values based on the known laws for cellular materials. Thanks to the use of different combinations of seed points and relaxation times, we were able to derive a relationship between Young’s modulus and the slope. In this way we were able to estimate the change of performance expected from the change of the type of resin.

Amendments in pages 10-11: “In order to check if the results shown… same scheme introduced in expression (11).”

+ three references added :

1.                  Dulieu-Barton, J.M.; Fulton, M.C. Mechanical properties of a typical stereolithography resin. Strain 2000, 36, 81-87.

2.                  Hague, R.; Mansour, S.; Saleh, N.; Harris, R. Materials analysis of stereolithography resins for use in rapid manufacturing. Journal of Materials Science 2004, 39, 2457-2464.

3.                  Quintana, R.; Choi, J.-W.; Puebla, K.; Wicker, R. Effects of build orientation on tensile strength for stereolithography-manufactured astm d-638 type i specimens. The International Journal of Advanced Manufacturing Technology 2009, 46, 201-215.

3)      Only 2D patterns have been discussed in the paper? What will happen if more complicated 3D patterns of cells were involved?

We extended the approach to three dimensional structures. We added the sub-section 4.2 to discuss the main achievements.

Major amendment in pages 11-12: section 4.2. Extension to 3D Voronoi/Monte Carlo scheme + Figures 10 and 11. + slight change in the conclusion section.

 Correspondence regarding this manuscript should be addressed to me at the address shown below.

Yours sincerely

Dr./ DHr.  Sofiane GUESSASMA

INRA

Rue de la géraudière

44316 Nantes, France

sofiane.guessasma@inra.fr

Reviewer 2 Report

Combing of Voronoi Diagrams and Monte Carlo Simualtions for optimisation of the Design for 3D Printing is not new.

) Can the Authors more explicitely point out the advantages of their combination ? Please acknowledge additional Groups working on this scientific topics.

    2.) Will the Programs be avaliabel for download and use?

    3.) Spell check required

Author Response

Unité de Recherche

Biopolymères, Interactions, Assemblages

Nantes, June 14th 2019

Dear Editor,

You will find enclosed our revised manuscript entitled « Designing cellular structures for additive manufacturing using Voronoi-Monte Carlo approach »

and written by

Tao Liu, Sofiane Guessasma, Jihong Zhu, Weihong Zhang

 We would like to take this opportunity to thank the editor and the reviewers for their comments and suggestions that all helped to elaborate a new strong version of this work. You will find hereafter a detailed answer to all these comments. All changes are also identified in the manuscript in yellow colour. 

  Reviewer 2 :

Comments and Suggestions for Authors

 Combing of Voronoi Diagrams and Monte Carlo Simualtions for optimisation of the Design for 3D Printing is not new.

 ) Can the Authors more explicitely point out the advantages of their combination ? Please acknowledge additional Groups working on this scientific topics.

The main advantages are introduced in the revised manuscript in the conclusion section at the light of the achieved results. Acknowledgement of research groups that worked in the Monte Carlo and Voronoi generation scheme is added in the introduction section.

Here is a list of the references that were cited in the revised manuscript.

Giannitelli, S.M.; Accoto, D.; Trombetta, M.; Rainer, A. Current trends in the design of scaffolds for computer-aided tissue engineering. Acta Biomaterialia 2014, 10, 580-594.

Bargmann, S.; Klusemann, B.; Markmann, J.; Schnabel, J.E.; Schneider, K.; Soyarslan, C.; Wilmers, J. Generation of 3d representative volume elements for heterogeneous materials: A review. Progress in Materials Science 2018, 96, 322-384.

Lee, M.; Fang, Q.; Cho, Y.; Ryu, J.; Liu, L.; Kim, D.-S. Support-free hollowing for 3d printing via voronoi diagram of ellipses. Computer-Aided Design 2018, 101, 23-36.

Rodgers, T.M.; Madison, J.D.; Tikare, V. Simulation of metal additive manufacturing microstructures using kinetic monte carlo. Computational Materials Science 2017, 135, 78-89.

Amendment in conclusion section: “The main advantage of…  number of Monte Carlo steps.”. + introduction section pages 2-3: “Indeed, according to the review… controlled connectivity between”

    2.) Will the Programs be avaliabel for download and use?

Up to now the authors are working on a user-friendly program with a graphical interface that will be available in the near future. Because of the substantial amount of time used to define the development environment, the link to this program is not yet available.

    3.) Spell check required

We checked the manuscript against lingual errors. All changes are also marked in the revised version.

 Correspondence regarding this manuscript should be addressed to me at the address shown below.

Yours sincerely

Dr./ DHr.  Sofiane GUESSASMA

INRA

Rue de la géraudière

44316 Nantes, France

sofiane.guessasma@inra.fr

Round  2

Reviewer 2 Report

 minor language corrections